# Influence of Blanching Time on Moisture, Sugars, Protein, and Processing Recovery of Sweet Corn Kernels

**Mariusz Szymanek [1,\*], Agata Dziwulska-Hunek [2]  and Wojciech Tanaś [1]**

[1] Department of Machine Science, University of Life Sciences in Lublin, 20-612 Lublin, Poland; wojciech.tanas@up.lublin.pl

[2] Department of Biophysics, University of Life Sciences in Lublin, 20-950 Lublin, Poland; agata.dziwulska-hunek@up.lublin.pl

\* Correspondence: mariusz.szymanek@up.lublin.pl; Tel.: +48-81-531-97-37

**Abstract:** The objective of this work is to determine the influence of blanching time on moisture content, sugars, protein, and processing recovery (degree of cut corn) of sweet corn kernels. Sweetcorn cobs of the Jubilee variety were blanched in the water at the temperature 85 °C during the period of 2, 4, 6, and 8 min. Non-blanched cobs were used as control samples. Blanched cobs were made subject to a mechanical process of cutting-off the kernels. It was stated that the blanching time has significant statistical influence on the content of moisture, sugars, and protein in kernel as well as on the quantity of the cut-off kernel mass.

**Keywords:** sweetcorn; blanching; moisture; sugars; protein; processing recovery

## 1. Introduction

Sweet corn (*Zea mays* (L.) var. saccharate) is a cultivated plant for human consumption, and it is a raw or processed material of the food industry throughout the world [1,2].

The process of a mechanical cutting-off the sweetcorn kernel from the cobs for consumer purposes (tinning, freezing) in food processing plants takes place with the help of classical cutting-off devices. Mechanical impact of knives upon the material being cut-off characterized by high moisture; above 70% causes great loss in quantity and quality including valuable nutrients [3]. The loss is generated not only at cutting-off but also as a result of further processing of the kernel (rinsing, blanching) during its contact with water [4]. Blanching is one of the stages of kernel technological production processes for consumer purposes. This process significantly affects the quality of the material [5]. The main purpose of blanching is the inactivation of the tissue enzymes responsible for spoiling the material as well as undesirable changes in its color, taste and texture. However, improper realization of this operation may cause the change of texture, color, the content of nutrients as well as affect the costs of the whole process [6]. Moreover, blanching influences the change of chemical and physical properties [7], not to mention the coagulation of the kernel pulp [8]. Changing the consistency of the grain pulp reduces the leakage of juice, and thus the loss of nutrients [9]. The changes which are created in the blanched material depend on the time duration, temperature, and the method of blanching [9] as well as on the variation, maturity and size of the corn cob [10]. The quality of the kernels also depends on whether the blanched occurs before the cutting process (corn on the cob) or after cutting it off. Boyet et al. [11] state that blanching the kernel on the cob, due to a wider area for blanching, results in deterioration of natural texture and taste of the kernel, whereas Peters and Ramirez [12] present that the blanched kernel before the cutting process contains a greater quantity of dimethyl sulfide than the blanched kernel after the cutting off. Trongpanich et al. [4] observed that blanching the cut-off kernel shows more

advantages than blanching the kernel on the cob before the cutting-off process. Higher properties for the yield of kernel, participation of solid bodies and nutrients were observed in the research. However, the authors stated that blanching the kernel on the cob before the cutting-off process results in the decrease of heat input used in the production process of the kernel. Niedziółka et al. [13] reported that blanching of cobs influences not only the quantity of the cut-off kernel but also its quality. In practice, very few operators blanch corn on the cob but blanch cut kernels because of the high energy requirements for ear blanching.

The objective of this work is to determine the influence of blanching time on moisture content, sugars, protein, and processing recovery (degree of cut corn) of sweet corn kernels.

## 2. Material and Methods

Sweet corn supersweet hybrid (Jubilee) was used in this study. This study was conducted during 2018 and 2019 at the field research of the University of Life Sciences in Lublin, Poland. The cobs for the research were collected manually during their processing maturity period defined on the basis of moisture and the consistence of the kernel pulp. Due to the fact that the blanching time depends on, among other things, the size of cobs, the research was therefore made on cobs having similar geometrical size.

The characteristics of the tested material are presented in Table 1.

**Table 1.** The characteristics of the tested material.

| Specification | Mean Value (SD) |
| --- | --- |
| Cob weight * (g) | 334.3 (21.1) |
| Cob length (cm) | 23.1 (2.6) |
| Cob diameter ** (cm) | 4.8 (0.8) |
| Number of kernels per row (pcs.) | 26 (3.2) |
| Number of kernel rows (pcs.) | 14.0 (2.4) |

* Husked cobs; ** measured at the central part of cob.

Blanching was carried out in a laboratory, blanching LW-11 (DanLab, PL) with 11 dm$^3$ capacity of water, heated by stainless steel electric resistance elements with a Pt100 probe for temperature control (Lumel RG24, PL).

Ten sweetcorn cobs were placed in a metal basket and immersed in a container with water of 85 °C. The blanching time was: 2, 4, 6, and 8 min. The conditions of blanching were chosen on the basis of the literature [14]. Cobs that did not undergo the blanching process (0) constituted the control set. After taking out the cobs from the container, they were cooled down in a flow of cold water until they reached temperature ca. 50 °C. Then they were dried. Blanching was carried out in three repetitions for each duration of time. Then the blanched cobs were submitted to the process of kernel separation in a cutter made by FMC model 3-AR. The angular speed of the cutter head was 167.5 rad·s$^{-1}$ and the linear velocity of the cob feeder was 0.31 m·s$^{-1}$.

The moisture content of the sweet corn kernels sample was determined gravimetrically by oven drying the sample at 103 °C for 72 h until a constant weight was obtained. The result was expressed as percent moisture loss (wet basis). The moisture of corn kernels was determined using the ASAE [15] method. The laboratory dryer KBC G-65/250(PREMED, PL) was used. The samples were allowed to cool in a desiccator, after which the weights were recorded. The weight was measured using a scale WPE 2000p (RADWAG, PL). Three samples were used and the average moisture content was reported.

Measurement of sugar contents was made using the DNS method following acid hydrolysis [16]. The reducing sugars were determined prior to the hydrolysis by means of the DNS method. The starch content was determined through the difference between the total content of sugars and the content of sugars soluble in ethyl alcohol (40% *v/v*). The sucrose content was established through the difference between the content of sugars soluble in ethyl alcohol and the content of reducing sugars. To determine

the reducing sugar content with the DNS method, a test tube was filled with 0.5 cm$^3$ of the tested solution and 1.5 cm$^3$ of 3-5-dinitrosalicylic acid (DNS) and then boiled for 5 min in a bath of boiling water. After cooling down, 6 cm$^3$ of distilled water was added to make a total volume of 8 cm$^3$. Next, the sample extinction was read against a reagent assay at a wavelength of λ = 550 nm. The extinction measurement results were referenced to the model curve. The sugar content was determined in relation to the kernel dry mass. The experiments were replicated thrice in 100-g samples for each variety and the average values were reported.

Protein content of kernel samples from each treatment were dried in an oven (65 °C) until obtaining constant mass. Afterward, total nitrogen was determined for each kernel sample according to [17] and the results expressed as proposed in the Kjeldahl method where protein content is estimated based on total nitrogen content.

The moisture of kernel and the content of sugars and starch were defined on the basis of 100-g samples which were taken at random from the pulp of the cut-off kernel. The end value of a given volume was determined by arithmetic average from three repetitions.

The degree of cutting off the kernel from cob cores was calculated on the basis of a percentage participation of the cut off kernel weight o the corn cob weight.

The statistical analysis of the research results was carried out by a method of a one-factor variance analysis and Tukey's multiple confidence intervals at an assumed significance level of α = 0.05.

## 3. Results and Discussion

From Figure 1, we can compare the moisture of the kernel corn from the different time of blanching. Moisture content of sweet corn during blanching varied with the blanching time. Mean values of moisture of the blanched cobs, within the duration of 2 to 8 min, were within the range from 77.4% to 78.6%. In relation to the non-blanched cobs with the moisture 76.7%, there occurred an increase in moisture of the blanched kernel by ca. 0.9% for 2 min, by ca. 1.2% for 4 min, by ca. 2.1% for 6 min, and by ca. 2.5% for 8 min of the blanching time. Tukey's multiple comparison tests showed, on the other hand, that mean values of moisture were significantly differentiated statistically only between the control (0) and 6 min blanching time and control (0) and 8 min blanching time.

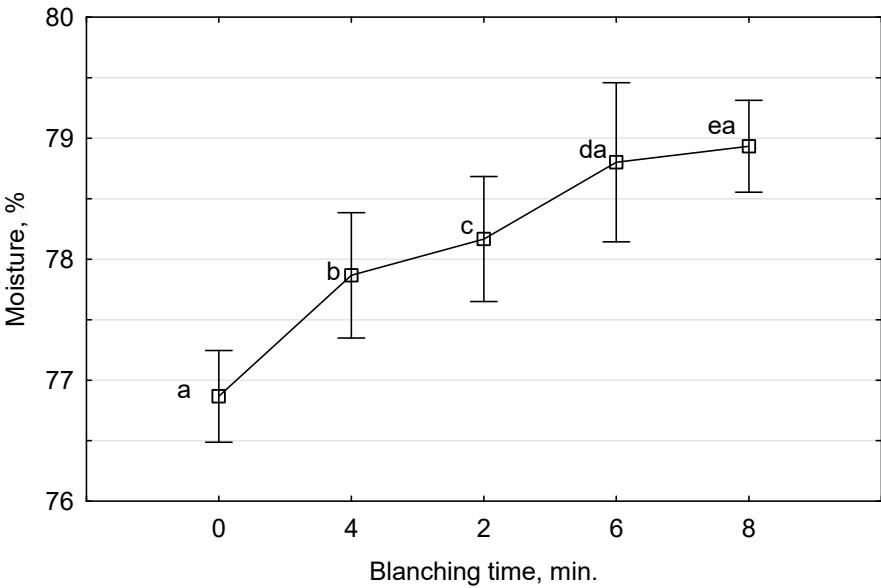

**Figure 1.** The influence of the blanching time on kernel moisture including 95% confidence intervals. Means with the same letter are significantly different from each other ($p < 0.05$).

Kachhadiya et al. [18] observed similar relationships while blanching sweet corn kernels. Moisture content significantly increased from 76% to 80.2% in hot water blanching.

Similar occurrences were observed in the research of other authors [19,20]. Kachhadiva et al. [18] reported, on the other hand, that in the case of some products, a significant decrease of moisture occurs only after a long-lasting blanching process, whereas making blanching time longer influences an increase of moisture.

Total sugar content of untreated sweet corn is extremely important and its content depends on variety, harvesting time, and post-harvest practices [21]. The analysis of sugar content in kernel showed a reverse process as for moisture. In relation to the content of sugars in non-blanched kernel (3.73 g), its content decreased to 2.90 g after 2 min, to 2.85 g after 4 min, to 2.82 g after 6 min, and 2.77 g after 8 min of the blanching time (Figure 2).

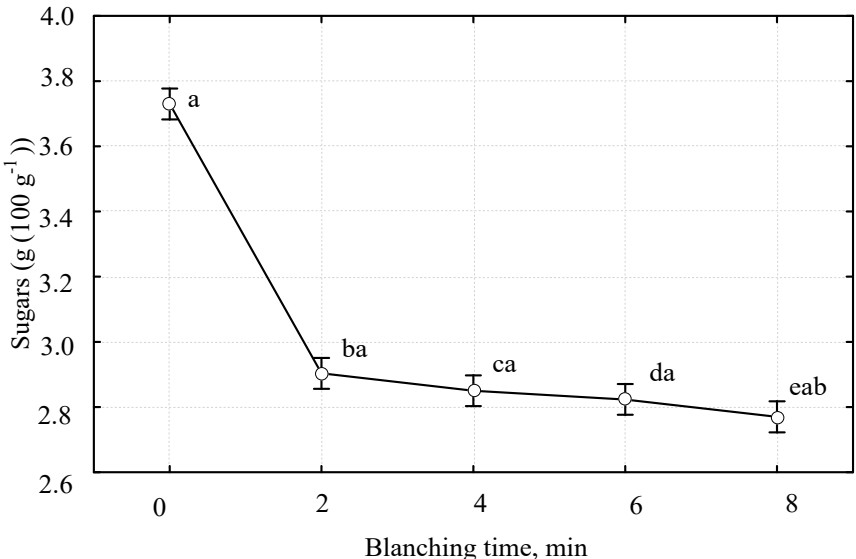

**Figure 2.** The influence of the blanching time on sugar content including 95% confidence intervals. Means with the same letter are significantly different from each other ($p < 0.05$).

Tukey's test showed, however, that mean values of sugars in blanched kernel were indeed only statistically between 2 and 8 min of blanching time. The decrease of the content of sugars in blanched material is proven by the research of other authors [22,23]. However, in the research of Califano and Calvelo [23], blanching was reported to be a process which minimizes the loss of sugars. Dauda [24] indicates that loss of soluble sugars from fresh corn is the major factor that results in staleness of fresh corn. Kachhadiya et al. [18] observed that total sugar significantly decreased from 8.40 to 5.70 g in hot water blanching. These results are in line with finding obtained by Alan et al. [25]. According to Shu [26], a decrease in sugar may be due to the leaching of sugar during blanching.

Mean content of protein in non-blanched kernel was 3.0 g. As the result of blanching, it decreased to 2.96 g after 2 min, to 2.81 g after 4 min, to 2.74 g after 6 min, and 2.71 g after 8 min of the blanching time (Figure 3). Tukey's test showed that significant statistical differences in decreasing the content of protein occurred only after 6 and 8 min of blanching time.

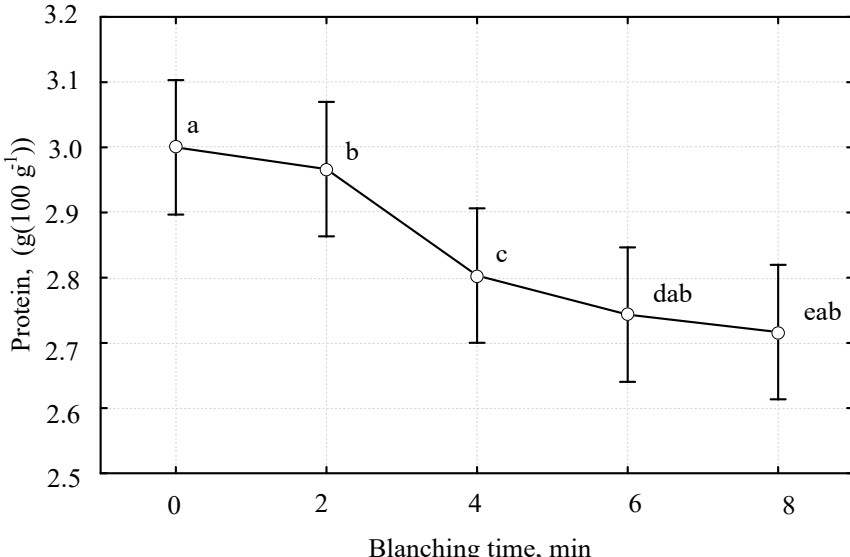

**Figure 3.** The influence of the blanching time on protein content including 95% confidence intervals. Means with the same letter are not significantly different from each other ($p < 0.05$).

Approximately a decrease of 7% of the protein content in spinach, which was submitted to ca. 3 min blanching in temperature about 85 °C, was noted by Grzeszczuk et al. [20]. A similar relationship is also shown in the work of other authors [27].

Mean value of the degree of cutting off the kernel mass from non-blanched was 56.26%. Together with making the blanching time longer, a significant statistical increase of cut-off kernel mass was observed. Mean values were 60.36% after 2 min, 70.07% after 4 min, 78.63% after 6 min, and 86.38% after 8 min of blanching (Figure 4).

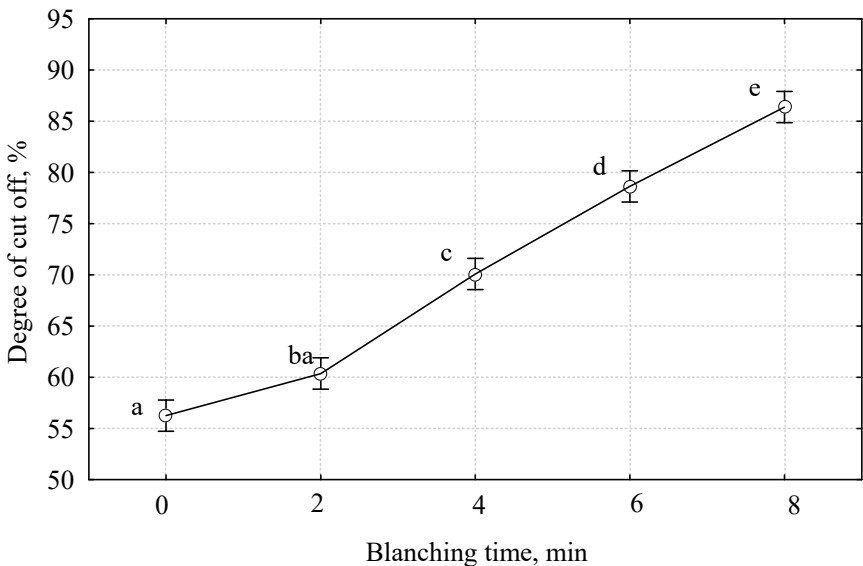

**Figure 4.** The influence of the blanching time on the degree of cut-off kernel including 95% confidence intervals. Means with the same letter are not significantly different from each other ($p < 0.05$).

The analysis of results showed that, similar to the research of Niedziółka et al. [13], blanching influences the increase of quantity of the cut-off kernel. However, this relationship was not stated by Trongpanich et al. [4]. From the course of events presented at Figure 4, it also results in blanching time significantly and statistically influencing the quantity of cut-off kernel mass. The increase of the cut-off kernel mass together with making blanching time longer can explain the change in moisture

and consistency of pulp (coagulation). Both these factors cause less loss of the kernel pulp during the kernel cutting.

## 4. Conclusions

1. Blanching time of sweet corn cobs has significant statistical influence on the content of moisture, sugars, and protein in kernel as well as on the quantity of the cut-off kernel mass.
2. The content of moisture and the degree of cutting-off increased, and sugars (in general) and protein decreased with the duration of time of the blanching process. The biggest mean values were obtained for 8 min blanching time. In relation to non-blanched kernel, the increase of moisture was observed up to ca. 2.%, the decrease of the content of sugars up to ca. 26%, and the content of protein up to ca. 10%, as well as the increase of degree of cutting-off the kernel pulp up to ca. 54%.
3. Blanching is extremely important for further processing sweet corn and extending shelf-life. In this study, various water blanching time were considered. The best blanching treatment for sweet corn based on the content of moisture, sugars, protein moisture content, and the cut-off kernel mass is 8 min.

**Author Contributions:** M.S., W.T. writing—review and editing, M.S., A.D.-H. conceived and planned the experiments., carried out the experiments. All authors have read and agreed to the published version of the manuscript.

**Funding:** This research was funded by The University of Life Sciences in Lublin.

**Conflicts of Interest:** The authors declare no conflict of interest.

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
