# Peer review of "Influence of Blanching Time on Moisture, Sugars, Protein, and Processing Recovery of Sweet Corn Kernels"

_processes, doi:10.3390/pr8030340_

Round 1

Reviewer 1 Report

This manuscript reports the effects of blanching time on moisture, sugars, protein and processing recovery of sweet corn kernels. The blanching period of 2,4,6 and 8 minutes were selected. It was suggested that the blanching time significantly affects the content of moisture, sugars and proteins in kernels. This finding is useful for scientists especially in the field of chemistry and food science. The following comments should be addressed before the paper can be further considered.

  • In the section of result and discussion, the authors should mention why the blanching duration of 2 to 8 min were selected for this study.
  • In the section of result and discussion, the authors should talk about the effects of moisture, sugars, protein and processing recovery of sweet corn kernels on human health based on their result.
  • In figure 1, 2, 3 and 4, the letters “a”, “b”, “c”, “d” and “e” should be clearly stated in figure caption. Do they mean statistical differences between groups? Which group are they comparing to?

Reviewer 2 Report

The article entitled "Influence of Blanching Time on Moisture, Sugars, Protein and Processing Recovery of Sweet Corn Kernels" is well structured, presenting suitable results and discussion.

Authors should closely revised english language: eg. line 22 should be "a plant cultivated for human consumption" or "a plant grown for human consumption); eg. line 27 should be better as "[…] kernel pulp causes the decrease of the juice effluent leading to the los of costly nutrients"; eg. line 40 would be more suitable as "the quality of the product blanched also depends on whether the kernel has been blanched before the cutting process (kernel on the cob) or after cutting of the cob"; etc.

Although references are provided for the methods carried out for the analysis of moisture, sugars and proteins, this reviewer suggest to additionally include a brief description of the procedure performed in each case.

Finally, I suggest the authors to include a conclusion about which blanching time would be the most suitable for sweet corn kernels according to their results obtained. 
